# Analysis of Genetic and MRI Changes, Blood Markers, and Risk Factors in a Twin Pair Discordant of Progressive Supranuclear Palsy

**DOI:** 10.3390/medicina59101696

**Published:** 2023-09-22

**Authors:** Aliz Persely, Beatrix Beszedics, Krisztina Paloczi, Marton Piroska, Amirreza Alijanpourotaghsara, David Strelnikov, Arsalan Vessal, Helga Szabo, Anita Hernyes, Luca Zoldi, Zsofia Jokkel, Andrea Fekete, Janos Juhasz, Nora Makra, Dora Szabo, Edit Buzas, Adam Domonkos Tarnoki, David Laszlo Tarnoki

**Affiliations:** 1Medical Imaging Centre, Semmelweis University, 1082 Budapest, Hungary; aliz.persely@gmail.com (A.P.); beszedics.beatrix@gmail.com (B.B.); piroskamarton94@gmail.com (M.P.); amirrezaalijanpour1373@gmail.com (A.A.); david.strelnikov17@gmail.com (D.S.); vessalarsalan@gmail.com (A.V.); szabo.helga.se@gmail.com (H.S.); anitahernyes@hotmail.com (A.H.); luca.zoeldi@gmail.com (L.Z.); zsofijokkel@gmail.com (Z.J.); fekete.andrea@med.semmelweis-univ.hu (A.F.); tarnoki2@gmail.com (A.D.T.); 2Neurology Department, Medical Centre Hungarian Defence Forces, 1134 Budapest, Hungary; 3Department of Genetics, Cell- and Immunobiology, Semmelweis University, 1085 Budapest, Hungary; paloczi.krisztina@med.semmelweis-univ.hu (K.P.); buzas.edit@med.semmelweis-univ.hu (E.B.); 4Central Radiological Diagnostic Department, Medical Centre Hungarian Defence Forces, 1134 Budapest, Hungary; 5Institute of Medical Microbiology, Semmelweis University, 1085 Budapest, Hungary; juhaszjanos4@gmail.com (J.J.); nora.makra@gmail.com (N.M.); szabo.dora@med.semmelweis-univ.hu (D.S.); 6Faculty of Information Technology and Bionics, Pazmany Peter Catholic University, 1085 Budapest, Hungary

**Keywords:** plasma biomarker, alpha diversity, dementia, risk factor, magnetic resonance, genetics, environment

## Abstract

*Background and Objectives*: Progressive supranuclear palsy (PSP) is a neurodegenerative disease, a tauopathy, which results in a wide clinical spectrum of neurological symptoms. The diagnosis is mostly based on clinical signs and neuroimaging; however, possible biomarkers for screening have been under investigation, and the role of the gut microbiome is unknown. The aim of our study was to identify potential blood biomarkers and observe variations in the gut microbiome within a PSP discordant monozygotic twin pair. *Materials and Methods*: Anthropometric measurements, neuropsychological tests, and the neurological state were evaluated. Blood was collected for metabolic profiling and for the detection of neurodegenerative and vascular biomarkers. Both the gut microbiome and brain MRI results were thoroughly examined. *Results*: We found a relevant difference between alpha-synuclein levels and moderate difference in the levels of MMP-2, MB, Apo-A1, Apo-CIII, and Apo-H. With respect to the ratios, a small difference was observed for ApoA1/SAA and ApoB/ApoA1. Using a microbiome analysis, we also discovered a relative dysbiosis, and the MRI results revealed midbrain and frontoparietal cortical atrophy along with a reduction in overall brain volumes and an increase in white matter lesions in the affected twin. *Conclusions*: We observed significant differences between the unaffected and affected twins in some risk factors and blood biomarkers, along with disparities in the gut microbiome. Additionally, we detected abnormalities in brain MRI results and alterations in cognitive functions.

## 1. Introduction

Progressive supranuclear palsy (PSP), also known as Steele–Richardson–Olszewski syndrome, is a neurodegenerative disease that is characterized by a wide clinical spectrum of symptoms, including gait and balance impairment, postural instability, ocular movement abnormalities (typically vertical supranuclear gaze palsy), facial and cervical dystonia, severe generalized bradykinesia, frontal dementia, visual impairment from gaze palsy, spastic/ataxic dysarthria, and dysphagia leading to aspiration. Less specific or inconsistent features include depression, sleep disturbance, urinary incontinence, constipation, apraxia, tremor, dystonia, and retrocollis.

The population prevalence for living cases in a community-based series approximates to about 5–6 cases per 100,000. Due to diagnostic difficulties, the incidence of PSP is estimated at 1–2 in 100,000 [1]. Survival from the time of symptom onset is around 7.4 years [1]. PSP affects patients over the age of 40 years with an average onset in the mid-60s with supranuclear gaze palsy or postural instability and consequential falls within the first year of the onset.

In terms of its neuropathology, PSP is a tauopathy. Tau is a microtubule-associated protein that provides stability for the axonal cytoskeleton. These proteins are genetically determined by a single gene located on chromosome 17 [2]. Through an alternative splicing mechanism, six isoforms of tau exist in the human brain. They are distinguished by the inclusion or exclusion of the region coded by exon 10 results, and three or four repeats are present. Due to certain abnormal posttranslational modifications, such as hyperphosphorylation, tau forms neurofibrillary tangles. The insoluble tau form is not exclusively composed of 4R and 3R forms, but there are less of 3R than 4R; however, in Alzheimer’s disease, this ratio is reversed. It is noteworthy to mention corticobasal degeneration (CBD) as it is also 4R tauopathy, and they are differentiated by their pathological features; PSP typically creates tufted astrocytes and globose neurofibrillary tangles, whereas CBD has astrocytic and ballooned pale neurons and more severely affects cortical regions. Furthermore, in PSP, oligodendroglia coiled bodies often appear in the thalamic and lenticular fasciculi [3].

The main neuroanatomical regions affected in PSP include the basal ganglia, subthalamic nucleus, substantia nigra, and frontal motor and premotor cortices, which show profound atrophy. The dentate nucleus and cerebellar outflow pathway are also usually affected, and atrophy of the superior cerebellar peduncle is also found [2,4]. Accordingly, brain magnetic resonance imaging (MRI) plays a valuable role in discriminating against atypical Parkinsonian syndromes, as radiological diagnoses seem to be more specific than clinical diagnoses [5]. Neuroimaging is an important tool for the diagnosis; the “hummingbird sign” is a common and traditional indicator of the disease. However, novel methods, such as volumetric analyses of MRI images, are great methods of assessment as well [6].

It has not yet been possible to identify a clear risk factor; mostly, environmental factors are known. However, genetic factors may also contribute to the development of the disease [7]. There are several potential risk factors that have been investigated, such as lower educational attainment, drinking well water, living near an agricultural area, and exposure to pesticides, which were shown to increase the risk of developing PSP [8]. Moreover, exposure to metals was also associated with PSP in a study examining veterans. In addition, the H1 haplotype of the microtubule-associated protein tau (MAPT) gene, which encodes tau protein, is the most significant genetic risk factor determined so far [9].

The gut microbiome is a key factor in the development and aging of the nervous system; therefore, the role of the gut–brain axis has already been investigated in neurodegenerative disorders. In most cases of the various neurodegenerative disorders, there were alterations; dysbiosis in the gut microbiome and a usually elevated pro-inflammatory state were observed. Altogether, it is known that with greater microbiome diversity, we can obtain better outcomes with respect to health [10,11]. However, the microbiome of PSP patients has not been profoundly researched so far.

Our aim was to find potential blood biomarkers in PSP while taking into account the brain MRI changes in a clinically discordant identical twin pair for PSP. This relationship has not been studied in twins as of yet.

## 2. Materials and Methods

### 2.1. Study Participants

We contacted 10,007 subjects in the population-based Hungarian Twin Registry [12] were to find twin pairs with PSP. We found a 76-year-old female monozygotic twin pair; they were involved in the study and were discordant for PSP. The twin pair had no prior carotid surgery, acute infection within three weeks of the study, underlying oncologic disease, inflammatory bowel disease, nor any acute respiratory, cardiac, and renal failure at the time of the study. The study was approved by the ethical committee (Semmelweis University TUKEB 217/2021). Both participants signed an informed consent form. The tenets of the Declaration of Helsinki were followed. Zygosity classification was determined using a seven-part, self-reported questionnaire [13]. Information about history and risk factors was obtained using a questionnaire, including height, body weight, body mass index (BMI), smoking, hypertension, hyperlipidemia, and diabetes. The examinations were performed at the Semmelweis University Medical Imaging Centre’s Department of Neuroradiology in Budapest, Hungary. 

### 2.2. Questionnaires

Personal interview-associated questionnaires and self-assessed questionnaires were completed to determine symptoms and comorbidities such as hypertension, cardiovascular disease, diabetes, dyslipidemia, and other important influencing factors such as smoking. Along with measuring body weight and height, BMI was also computed.

Among the neurocognitive and psychological tests, we assessed the neurocognitive functions of the participants using the Montreal Cognitive Assessment (MoCA), Addenbrooke Cognitive Examination (ACE), and Mini-Mental State Examination (MMSE) tests; then, the mental health of the patients was evaluated using the Beck Depression Inventory (BDI), Zung Self-Rating Depression Scale (ZDS) and Geriatric Depression Scale (GDS). It should be considered that ACE, MMSE, and MoCA have different cut-off scores in the literature for cognitive impairment; however, the higher the score, the better. If the MMSE score is below 24 points, the result is frequently thought to be abnormal. For ACE, 88 and 83 cut-off scores are usually recommended. If we find that the score of the MoCA test is below 25, a cognitive impairment might be present. The cut-off score for BDI is 9, GDS is 5, and ZDS is 41. [14,15,16,17]. 

The twin pair underwent a body composition analysis by body impedance analysis (BIA) using a clinically validated, portable body consistency monitor (OMRON BF500, Omron Healthcare Ltd., Kyoto, Japan) [18]. Body fat percentage was calculated as [body fat mass (kg)/body weight (kg)] × 100. Fat-free mass was interpreted as [100% − body fat percentage (%)]. Waist and hip circumferences were measured using standard criteria. Blood pressure was measured using a TensioMed Arteriograph (TensioMed Ltd., Budapest, Hungary).

### 2.3. Sample Collection

Venous blood was taken one time from the twin pair on the same day in the year 2022. Blood was collected in serum, EDTA, and ACD-A tubes. We used tubes containing a coagulation accelerator that was suitable for serum extraction. Plasma samples were prepared by centrifugation twice at 2500× *g* for 15 min at 22 °C (Eppendorf 5804R), aliquoted, and stored at −80 °C before analysis. 

### 2.4. Routine Laboratory Testing

Serum triglycerides, total cholesterol, low-density lipoprotein cholesterol (LDL cholesterol), apolipoprotein A1 (ApoA1), apolipoprotein B100 (ApoB), and lipoprotein (a) (Lp (a)) levels were determined in fasting state using a Beckman Coulter AU680 analyzer at the Institute of Laboratory Medicine, Semmelweis University.

### 2.5. Bead-Based Multiplex Immunoassay

Three bead-based multiplex assays were used to quantify apolipoprotein (Apo) proteins and neurodegenerative and vascular biomarkers. The Human Apo Panel allowed for the simultaneous quantification of 11 human Apo proteins, including Apo A1, AII, ApoB 100, CII, CIII, D, E, E4, H, J, and M. The Human Neurodegenerative Disease Biomarker Panel provided the detection of the native proteins α-synuclein (αSyn), tau (TAU), β-amyloid-40 (Aβ40), β-amyloid-42 (Aβ42), and neurofilament light chain (NFL). The Human Vascular Inflammation Panel-1 conducted a simultaneous quantification of 12 human proteins, including myoglobin (MB), calprotectin (MRP8/14), lipocalin A (NGAL), osteopontin (OPN), myeloperoxidase (MPO), serum amyloid A (SAA), insulin-like growth factor-binding protein-4 (IGFBP-4), intercellular adhesion molecule 1 (ICAM-1) (CD54), vascular cell adhesion protein 1 (VCAM-1) (CD106), *matrix metalloproteinase-2* (MMP-2), *matrix metalloproteinase-9* (MMP-9), and cystatin C (LEGENDplex, Biolegend, San Diego, CA, USA). Plasma samples were diluted at a ratio of 1:8000 for the Apo Panel, 1:2 for the Neurodegenerative Panel, and 1:100 for the Vascular Inflammation Panel-1. The standards and samples were measured in two replicates. All assays were performed in accordance with the manufacturer’s instructions. The signal intensity of microbeads was detected using a CytoFLEX instrument (Beckman Coulter, Brea, CA, USA). The gate setup included both A and B beads (gate A + B), and the number of events was adjusted according to the panel so that 300–600 events were recorded for each analyte. The files were analyzed using FlowJo v.X.0.7 software (Tree Star Inc., Ashland, OR, USA). A standard curve that was specific to the kit lot number was constructed for each analyte, and the data were analyzed using the GraphPad Prism 9 software (GraphPad Software Inc., San Diego, CA, USA); the final concentrations were obtained after correcting for sample dilution.

### 2.6. Genotyping Apolipoprotein E Gene

Genomic deoxyribonucleic acid (DNA) was isolated from EDTA blood samples using a Tissue/Blood DNA Mini Kit [Geneaid; GB300]. Three allele-specific real-time PCR (qPCR) reactions were performed to investigate the APOE gene polymorphisms (ε2/ε2, ε2/ε3, ε2/ε4, ε3/ε3, ε3/ε4, ε4/ε4). For all reactions (15 µL), 1× TaqMan^®^ Genotyping Master Mix (Thermo Scientific, Waltham, MA, USA), 0.5 μM of APOE primers, and an APOE assay probe were used as an internal control. Therefore, no repeats of genotyping were needed. A pair of 1 μM of beta actin (ACTB) primers and an ACTB assay probe were used (Merck) (Stable-1). A total of 5µL (~5 ng) of genomic DNA was used as the template. The first step of the PCR amplification protocol involved activation of AmpliTaq Gold DNA polymerase at 95 °C for 10 s, followed by 16 cycles of a touchdown PCR to achieve higher specificity; denaturation at 95 °C for 10 s and annealing/extension at 70 °C for 30 s, with each cycle reducing by 0.5 °C down to 62 °C; and denaturation at 95 °C for 10 s and annealing/extension at 62 °C for 30 s for 20 cycles (CFX C1000 Touch 96-well; Bio-Rad, Hercules, CA, USA). The evaluation took into account the presence of APOE FAM, and ACTB HEX RFU as well as the allele discrimination results [19] (Appendix A).

### 2.7. Stool Sample Collection and Processing, Bioinformatics, and Statistical Analysis

A standardized procedure [20] was followed for the collection and processing of the stool samples as well as for the bioinformatics and statistical analysis. The participants received a detailed instruction manual with a flow chart for the purpose of standardized sampling, in which the importance of the short time between sampling and return; the prevention of external contamination; and the quantity, proper storage, and packaging of the sample were highlighted. Samples were stored and frozen prior to processing. For the hypervariable region V3–V4 of microbial 16S ribosomal RNA, a library extraction was conducted after DNA extraction. Prior to sequencing on an Illumina MiSeq platform (Thermo Fischer Scientific, Waltham, MA, USA), the libraries were tagged with individual index pairs, verified using an Agilent 2100 Bioanalyzer (Agilent, Santa Clara, CA, USA), and pooled. 

Sequencing data were analyzed using the Nephele cloud platform (NIH, NIAID) (https://nephele.niaid.nih.gov/index (accessed on 21 June 2023)) [21]. Quality control and read preprocessing were performed on the platform using the default parameters. The QIIME2 pipeline [22] was used after quality control for next-generation sequencing (NGS) read classification with closed reference OUT clustering based on the SILVA database [23]. Default values were set for the other parameters. Shannon entropy was calculated to estimate the alpha-diversities of the samples and weighted UniFrac distance-based principal coordinate analysis (PCoA) was performed to evaluate the distance between their compositions in the principal component space. Stacked bars were used for comparing the relative abundances of the most common bacterial taxa of the two samples. Groups (in each taxonomic level) with relative abundancies of at least 4% in one of the samples or on average were included in these comparisons. 

### 2.8. MRI Acquisition

Brain MRI imaging was carried out in April 2022 on a Philips Ingenia 1.5 T scanner (Philips Healthcare, Best, The Netherlands) at the Semmelweis University Medical Imaging Centre in Budapest, Hungary. In addition, previous brain MRI images were also assessed, which were taken in February, 2015 on a Siemens Magnetom Verio 1.5 T scanner (Siemens Healthcare GmbH, Erlangen, Germany) at the Borsod County University Teaching Hospital in Miskolc; for the PSP twin, an additional brain MRI was performed in October, 2019 on a Philips Ingenia 1.5 T scanner (Philips Healthcare, Best, The Netherlands) at the Semmelweis University. T1-weighted (T1W), T2-weighted (T2W), trace-weighted diffusion, apparent diffusion coefficient, proton density, and T2* and T2W dark fluid (FLAIR) images of the brain were taken. There was no contrast agent used. To assess the white matter abnormalities (white matter hyperintensities, WMHs), we employed T1W and T2W dark fluid (FLAIR) images. The following imaging parameters were used in the Philips scanner: TE/TR 140/9000 ms, flip angle 88°, 290 × 336 × 336 matrices, 0.8333 × 0.8333 in-plane resolution, and 0.6 mm slice thickness. Both twin pairs were always scanned on the same scanner on the same day (except for 2019). 

### 2.9. Image Processing

The 3D T1-weighted images and FLAIR images, originally in DICOM (Digital Imaging and Communications in Medicine) format, were converted to NIfTI (Neuroimaging Informatics Technology Initiative; http://nifti.nimh.nih.gov/ (accessed on 18 January 2023) format using a DCM2NII converter (http://www.mricro.com (accessed on 18 January 2023), mricron; Chris Rorden, Columbia, SC, USA). This format was then employed for all subsequent image processing [24]. 

For white matter hyperintensity analysis, we used the volBrain’s DeepLesionBrain (DLB) pipeline (https://www.volbrain.upv.es (accessed on 8 February 2023)). VolBrain is an MRI brain volumetry software that operates automatically and can offer brain structure volumes without human involvement; it was developed by José V. Manjón (IBIME, UPV, Valencia, Spain) and Pierrick Coupé (LaBRI UMR 5800, Université de Bordeaux, CNRS, Bordeaux, France). VolBrain employs an entirely automated pipeline for volumetric brain analysis based on multi-atlas label fusion technology, which can provide accurate volumetric information at various levels of detail in a short time [25]. WMH segmentation starts with image denoising, which is followed by inhomogeneity correction, spatial registration, intensity normalization, and intracranial cavity extraction by employing the Montreal Neurological Institute algorithm (MNI). The tissue is then segmented using a multi-template fusion atlas strategy, originating from a library that was produced by manually segmenting 43 patients by an expert radiologist using multimodal MRI data. Voxels surpassing a specific threshold are lesion candidates; the volBrain program automatically handles thresholding and voxel processing. The identified lesions are allocated into four anatomical regions: paraventricular, deep white, juxtacortical, and infratentorial, which are further divided into cerebellar and medullar regions. The number, volume, and distribution of each lesion are recorded. The process concludes with the generation of an automated report containing the lesion load, number of lesions per class, and screenshots of the processed images. Additionally, DLB provides the probability of disconnection caused by the detected lesions for 64 white matter tracts and estimates a disconnectome map based on the HCP1065 atlas [26]. Figure 1, Figure 2 and Figure 3 demonstrate MRI FLAIR sequence segmentation using volBrain DLB.

## 3. Results

Table 1 displays the basic patient data of the twin pair. The affected twin was the firstborn twin. Both twins had a high school diploma, both performed intellectual work throughout their lives, and both were married. The unaffected twin lived in a close family, whereas the affected twin lived in a multigenerational family. Both stayed in a big city during their lives. No alcohol was consumed. Both twins were breastfed for over 6 months. 

However, remarkable differences were found between the two twins (as shown in Table 1). The affected twin did not regularly exercise and did not drink coffee but was a heavy smoker, suffered from diabetes and breast cancer, suffered from a COVID infection, and took contraceptives compared with her unaffected twin. The disease started to show signs about 3–4 years before our examinations according to the patient and her relatives, but there were no symptoms in the other twin as of yet. 

A body composition analysis demonstrated a lower body mass index, body fat percentage, and visceral fat in the affected twin (Table 2). In addition, the affected twin had elevated blood pressure, but no remarkable differences were found in the serum lipid levels between the twins. 

Considering the LegendPlex data, there was no relevant difference between the two members of the twin pair in the levels of Apo-D, Apo-M, OPN, Apo-J, Aβ42, NFL, ICAM-1, NGAL, MRP8/14, SAA, and Apo-AII. There was a small difference in the levels of MMP9, Aβ40, TAU, VCAM-1, CysC, Apo-B100, Apo-CII, and Apo-E; a moderate difference in the levels of MMP2, MB, Apo-A1, Apo-CIII, and Apo-H; and an extremely high level difference in the level of αSyn (Table 2). Regarding the ratios, there was no difference between the two members of the twin pair in the levels of TAU/Aβ42 and Aβ42/Aβ40 while there was a small difference in the levels of ApoA1/SAA and ApoB/ApoA1. There was no significant difference between the two members of the twin pair in the levels of triglyceride, total cholesterol, low-density lipoprotein cholesterol (LDL-C), high-density lipoprotein cholesterol (HDL-C), ApoB, ApoA1, and their ratio. However, it is known that there is a higher cross-reactivity in the LegendPlex analysis that affects Apo-A1 and Apo-B100; therefore, these particular parameters were determined traditionally using a routine fasting state method.

A neurological assessment was conducted on the affected twin, which revealed gait loss, vertical gaze palsy, consecutive visual impairment, facial dystonia, dysarthria, bradykinesia, rigor, and retrocollis. The patient also reported urinary incontinence and sleep disturbances. In contrast, the other twin did not exhibit any symptoms or complaints and was clinically negative for PSP or any other neurodegenerative disease.

Table 3 displays the results of the neuropsychological examination. According to these tests, the twin pair presented prominent differences in the ACE, MMSE, and MoCA test results. Conversely, when it came to mental health questionnaires such as BDI, GDS, and ZDS, there were no notable variations, and none indicated concern for depression. 

In the ACE analysis, we observed a decrease in total points indicating global cognitive deficiency and dementia. Additionally, there was a larger reduction in the sub-scores of attention, memory, fluency, and visuospatial skills. The MMSE result was also reduced in the affected twin, which indicated mild dementia. According to the MoCA test, a moderate cognitive impairment was present (Table 3).

Concerning neuroimaging, we detected a left cerebellar lacunar stroke and signs of PSP (midbrain atrophy) in the affected twin; signs of normal-pressure hydrocephalus (NPH) were also observed. However, the latest brain MRI certified questionable signs of a possible beginner PSP, a narrow oedematous signal disorder in the cerebellar tonsils, and possible signs of normal-pressure hydrocephalus in the unaffected twin (Figure 1 and Figure 2).

Based on the DLB analysis using the HCP1065 atlas, out of the 64 analyzed white matter tracts in the affected twin pair, 60 tracts were affected by lesions (disconnection probability of above 55%), excluding left and right fornices and the cingulum parahippocampal tracts. On the other hand, in the unaffected twin pair, more than 33% of the analyzed tracts were unaffected by lesions (disconnection probability of 0%).

The white matter hyperintensity analysis using volBrain’s DLB pipeline demonstrated remarkable differences between the twins, especially in the total, periventricular, and juxtacortical WMH lesion volume, which were higher in the affected twin (Table 4).

The brain volumetric analysis showed a remarkable decrease in brain volumes in the affected twin, mainly in total gray matter and frontal lobe volume, except for the temporal lobe volume (Table 5). 

The volumetric analysis of the cerebrospinal fluid (CSF) volume in the twins showed a noteworthy increase in the CSF volume of the affected twin, which was indicative of atrophy in multiple brain regions, and a pronounced dilation of the third and fourth ventricles (Table 6). 

Alpha diversity, which is a measure of the diversity of the microbiome in a single sample, was calculated, and the results showed that the affected twin had a smaller Shannon diversity (7.04) than the unaffected twin (8.51). Beta diversity, which provides more information about the alikeness or dissimilarity of more communities, was also evaluated. The principal coordinate analysis (PCoA) constructed using weighted UniFrac distances showed that the samples were not particularly similar based on their microbial composition (Figure 4).

The microbiome composition assessment was accomplished. The plots revealed the relative abundances (in %) of the most common taxa in the two samples (taxa occurring with at least 4% in one of the samples, or at least 4% on average). The lower plots showed the relative abundance differences (in %) between the unaffected and affected twin. 

At the phylum level, we found that *Firmicutes* was the dominant phylum in both samples. The unaffected twin had higher *Firmicutes* and *Proteobacteria* content, while the sample of the affected twin had more *Bacteroidota* and *Verrucomicrobiota*. (Figure 5).

We examined our microbiome data at the genus level as well and found that the unaffected twin’s sample had more bacteria belonging to the *Bacteroides*, *Anaerostipes* genera, *Lachnospiraceae* family, and *Enterobacteriales* order. The latter three were rare in the sample of the twin with the disease, wherein the sample had a much higher content of *Akkermansia*, *Oscillospiraceae*, and *Odoribacter*. (Figure 6).

## 4. Discussion

To the best of our knowledge, this is the first study analyzing the risk factors, genetic and biomarker changes, and gut microbiome in PSP in a twin pair. The identification of environmental and modifiable risk factors and potential new plasma biomarkers creates an opportunity to diagnose the condition early, intervene, and delay the onset of PSP or slow disease progression. Although the study analysis was only performed on one twin pair, who share nearly 100% of their genes, we found remarkable differences in lifestyle factors during their lifespan (e.g., lack of regular exercise, coffee consumption, and heavy smoking), blood neurodegenerative and vascular biomarkers (especially in case of αSyn, as well as MMP-2, MB, Apo-A1, Apo-CIII, Apo-H levels), neuropsychological tests (ACE, MoCA, MMSE), and brain MRI findings (cerebellar lacunar stroke, midbrain atrophy, signs of NPH, WMH lesions and brain volume changes) in the affected twin. Moreover, differences regarding the gut microbiome were also observed.

To date, few articles have focused on environmental and lifestyle risk factors for PSP. In the review of Park et al., exposures to toxins related to diet, metals, well water, and hypertension were associated with an increased risk of PSP, while higher education and statins were protective [27]. We found important differences by analyzing the risk factors of the twins during their lifespan. There was no difference in education, working conditions, marital status, social conditions, and breastfeeding. However, lack of regular exercise, regular coffee consumption, and heavy smoking was found in the affected twin, which was not yet known as a risk factor of PSP. Previous studies did not certify an association between PSP and smoking habits [27,28,29] except in the univariate analysis of the ENGENE-PSP study, which did not reach statistical significance in the multivariate analysis (OR = 1.10; 95% CI = 0.99–1.22; *p* = 0.08) [30]. The PSP-affected twin had hypertension, which correlates with the previous findings [27,31]. In addition, the affected twin had been living with diabetes for four years, previously underwent breast cancer surgery, experienced a COVID infection, and had taken contraceptives. 

A body composition analysis demonstrated a lower body mass index, body fat percentage, and visceral fat in the PSP-affected twin, which was associated with fatigue due to disease progression. 

No remarkable differences were found in the serum lipid levels between the twins, which correlates with a previous multicenter study [31]. 

Blood protein biomarkers are being increasingly used to reliably differentiate PSP from healthy controls and patients with relevant differential diagnoses [32,33]. Serum uric acid seems to be lower in PSP, whereas methyl malonate and homocysteine seem to be elevated [34]. According to Chouliaras et al. [32], patients with PSP have a high NFL but no significant increase in other biomarkers such as P-tau181, Aβ42/40, and GFAP (glial fibrillar acidic protein) [35], which was partially found in our case, as TAU and Aβ42/40 were decreased in the twin diagnosed with PSP. However, p-tau181 and GFAP were associated with baseline cognitive function in PSP [33]. Baseline NFL was found to be a predictor of disease progression in PSP. Considering the blood neurodegenerative and vascular biomarker findings, we found small, negligible differences in the levels of MMP-9, Aβ40, TAU, VCAM-1, CysC, Apo-B100, Apo-CII, and Apo-E; a moderate difference in the levels of MMP-2, MB, Apo-A1, Apo-CIII, and Apo-H; and an extremely high-level difference in the level of αSyn. There was a small difference in the levels of ApoA1/SAA and ApoB/ApoA1. The aforementioned biomarkers may become potential biomarkers if further investigations confirm it on a larger sample size.

Many studies have reported various structural neuroimaging findings in T1- weighted, T2-weighted, and FLAIR images, including midbrain atrophy, atrophy of the superior cerebellar peduncle, and frontal and parietal cortical atrophy, which were present in the affected twin. Moreover, dilatation of the third and fourth ventricles and the aqueduct of Sylvius was also observed, which was present in the twin pair, with dominance in the affected twin. In addition, atrophy of the subthalamic nucleus is the most typical trait of PSP. The most characteristic pathological finding is the atrophy of the midbrain, which is demonstrated by the hummingbird sign as a result of rostral midbrain atrophy observed on mid-sagittal images, which was observed in the affected twin [6,36]. Interestingly, MRI signs of possible beginner PSP were suspected in the MRI imaging for the unaffected twin despite them being clinically asymptomatic, including narrow edematous signal disorder in the cerebellar tonsils and possible signs of NPH. The WMH analysis demonstrated remarkable differences between the twins, especially in the total, periventricular, and juxtacortical WMH lesion volume, which were higher in the PSP-affected twin. In addition, a remarkable decrease in brain volumes, mainly in total gray matter and frontal lobe volume except for temporal lobe volume, was observed. In the affected twin, a left cerebellar lacunar stroke was also observed, which referred to a vascular (possibly atherosclerotic) origin.

Neurocognitive decline and dementia are common findings in PSP and often determine the individual’s quality of life. From a broad reduction in overall global cognition to a more specific frontal behavioral changes, symptoms such as decreased verbal fluency, impaired abstract thought, and reduced motor functions are observed in patients with PSP [37]. Apathy, dysphoria, and anxiety may also be present [38]. In our case study, the patient with PSP did not show any signs of depression; in contrast, she suffered from cognitive decline. Regarding the ACE test, the affected twin had a reduction in the cognitive domains of attention, memory, fluency, and visuospatial skills, which are well-known indicators of PSP. On the other hand, MMSE, the most broadly used screening method for the evaluation of cognitive impairment, only showed mild dementia, while the MoCA test implicated a moderate decline in cognition. Fiorenzato et al. [14] reflected that the MoCA test is more sufficient and sensitive in the workup of PSP than MMSE, meaning that our results are in agreement with their findings. The ACE test showed the classical diagnostic signs of lost functions with respect to cognition. 

The connection between the nervous system and the gut, known as the gut–brain axis, has been under investigation recently, especially regarding neurodegenerative disorders because of dementia’s incidence growing every day worldwide. In the last decade, several studies have been trying to explore the correlation between the gut microbiota and cognitive decline, and there has still not been a crystal clear explanation regarding this relationship; however, many studies suggest that gut microbiota can modulate brain function [10,39]. For example, Alzheimer’s disease (AD) was associated with higher *Akkermansia* and lower *Firmicutes* levels [40]. A reduced volume of *Lachnospiraceae* and *Rikenellaceae* family members was also correlated with AD [41,42]. In contrast, other studies showed similar *Lachnospiraceae* content in PSP patients compared with healthy controls [43]. The clinically affected twin’s sample has less *Lachnospiraceae* but more *Rikenellaceae* (*Alistipes*) than the asymptomatic twin. Tian et al. applied a fecal microbiome transplantation (FTM) on PSP-Richardson’s syndrome patients in order to reduce dysbiosis [44]. The authors reported improvement in the symptoms of the patients as well as reduced intestinal inflammation and a healthier intestinal barrier due to the increase of short-chain fatty acid (SCFA)-producing bacteria; they also reported more *Bacteroidota* and fewer *Firmicutes* in the PSP patients compared with the control ones and associated SCFA-producing taxa (e.g., *Faecalibacterium*, *Blautia*, *Roseburia*) for health as well. The healthy twin’s sample had a higher ratio of these SCFA-producing bacteria than the twin with PSP dementia. The PSP-affected twin had more *Oscillospiraceae* and *Odoribacter*, and these bacterial species have the potential for producing SCFAs; *Oscillospira* sp. can produce butyrate [45], whereas *Odoribacter laneus* can synthesize propionate by reducing succinate [46]. This result indicates that these bacteria can have anti-inflammatory effects, while the amount of them and the higher relative abundance of *Akkermansia* are all associated with the microbiome of older individuals [47]. Our results suggest that the microbiomes of this pair are notably different; the affected twin seemed to have older characteristics than the microbiome of the possibly healthy twin sibling. The affected twin was not seriously dysbiotic but showed some signs that were associated with AD and PSP (lower anti-inflammatory environment).

The main limitation is that this is a case study, and the findings must be considered with caution and cannot be generalized. Although we received similar data (especially observing the cognitive tests and MRI results) in comparison with the general population and with healthy and affected individuals, our sample amount was small; further investigations are important. Discordant twin examination was not possible as this was the only discordant MZ pair in the registry where PSP was present as a rare condition. A second limitation is that other existing conditions (such as vascular dementia), chronic diseases, and drug therapies in our subjects might also bias the results. 

## 5. Summary and Conclusions

In our single case, taking into account the study design, we found remarkable differences in some risk factors (less exercise, heavy smoking), in the level of blood neurodegenerative and vascular biomarkers (α-synuclein, MMP-2, MB, Apo-A1, Apo-CIII, Apo-H), and in the gut microbiome in PSP-related twin subjects, which was correlated with relevant neuropsychological tests and typical brain MRI abnormalities. Further investigations are necessary to reveal these findings in a larger sample, and explicit conclusions cannot be drawn. 

## Figures and Tables

**Figure 1 medicina-59-01696-f001:**
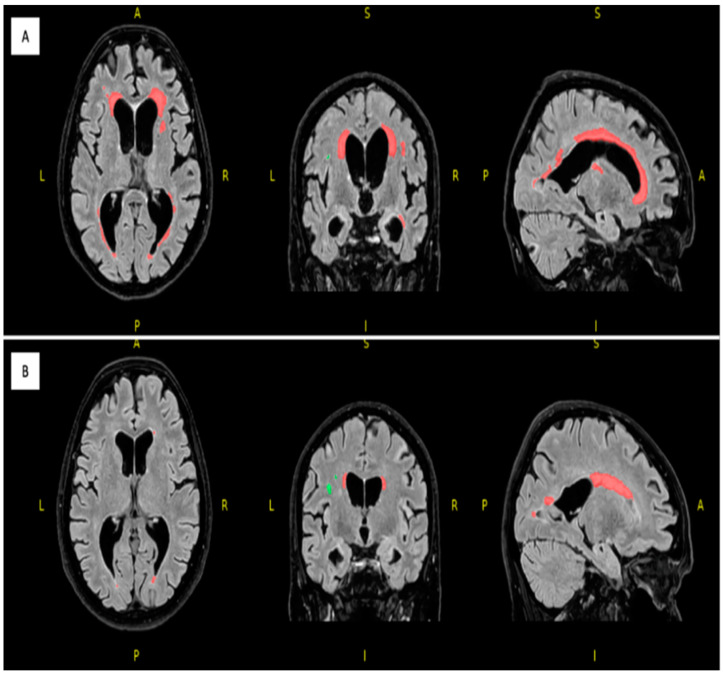
Segmented FLAIR MRI imaging of the 77 years old female monozygotic twin pair in 2022 using DLB (axial, coronal and sagittal planes); the affected twin (**A**) has a total number of 36 WMHs (first row) and the unaffected twin (**B**) has a total number of 36 WMHs (second row). Red represents the periventricular WMH and green shows the deep white matter WMH; image from the Semmelweis University Medical Imaging Centre. A: anterior; P: posterior; L: left; R: right.

**Figure 2 medicina-59-01696-f002:**
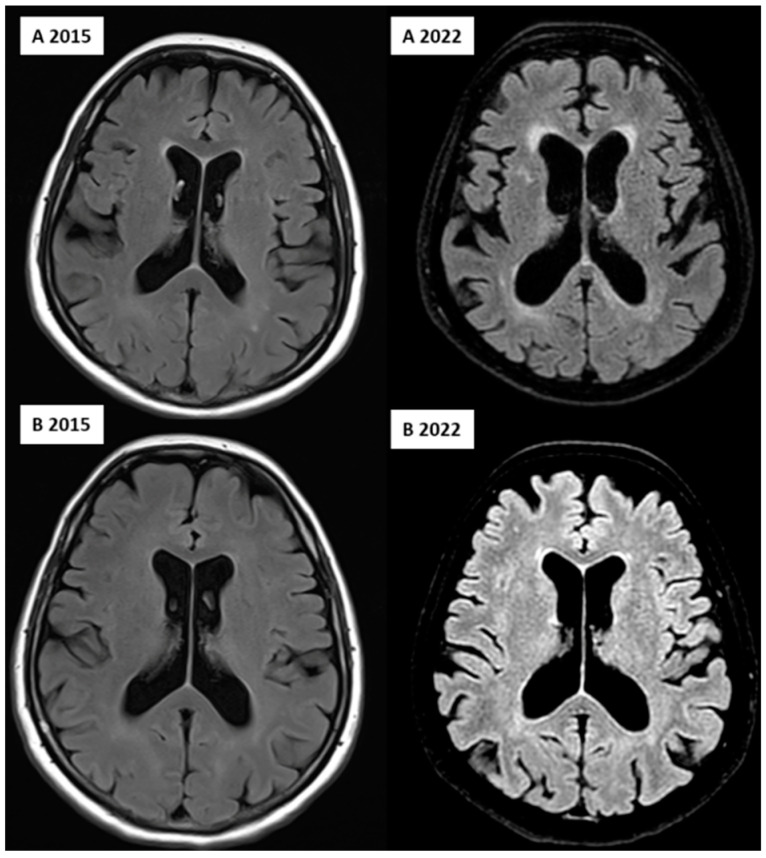
FLAIR image of the affected (**A**, **upper row**) and unaffected (**B**, **lower row**) twin over 7 years, showing the progression of the WMHs in the deep and periventricular regions and ventricular dilation, indicating global cerebral atrophy; image from the Semmelweis University Medical Imaging Centre.

**Figure 3 medicina-59-01696-f003:**
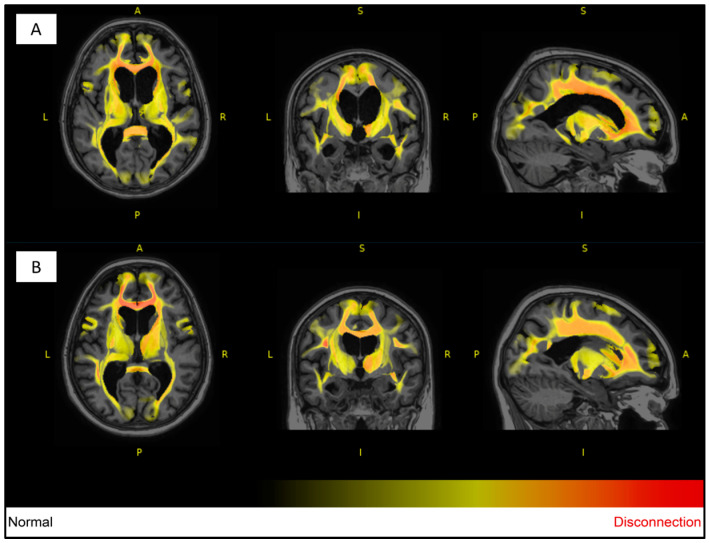
Segmented T1W MRI imaging of the 77-year-old female monozygotic twin pair in 2022 using DLB; most of the analyzed tracts of the affected twin (**A**) have been affected by lesions (first row), and more than a third of the analyzed tracts were completely unaffected by lesions in the unaffected pair (**B**) (second row). The degree of WMH impact on each tract is visualized through a heat map, with yellow and red indicating increasing severity. Image from the Semmelweis University Medical Imaging Centre. A: anterior; P: posterior; L: left; R: right.

**Figure 4 medicina-59-01696-f004:**
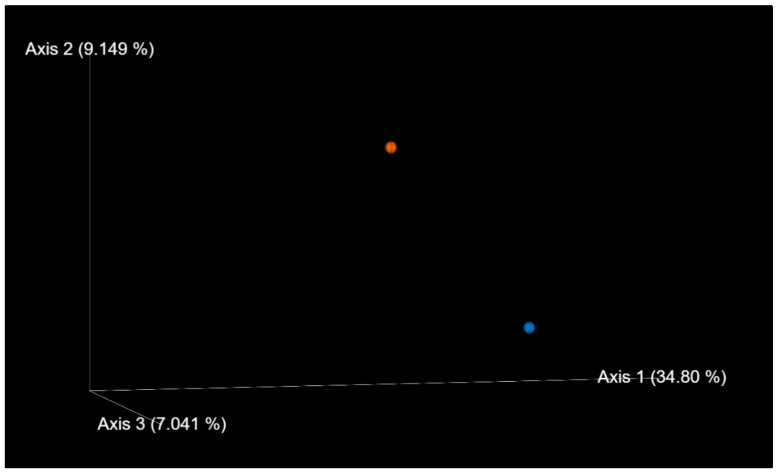
Positions of samples: unaffected (red) and affected (blue) samples on the weighted UniFrac distance-based principal coordinate space. Directions of the three principal components in the weighted UniFrac distance space (% values denote the percentages of variances explained by the axes).

**Figure 5 medicina-59-01696-f005:**
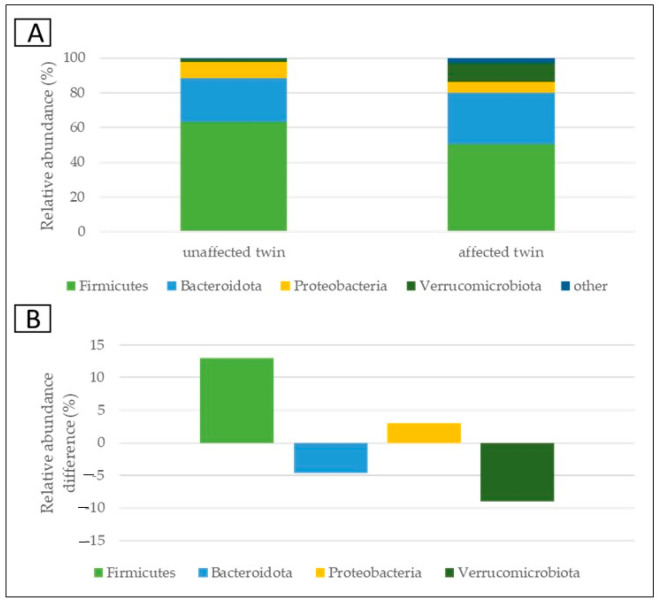
Relative abundances at the level of phylum (**A**); the differences between abundances with respect to the PSP discordant twin pair (**B**).

**Figure 6 medicina-59-01696-f006:**
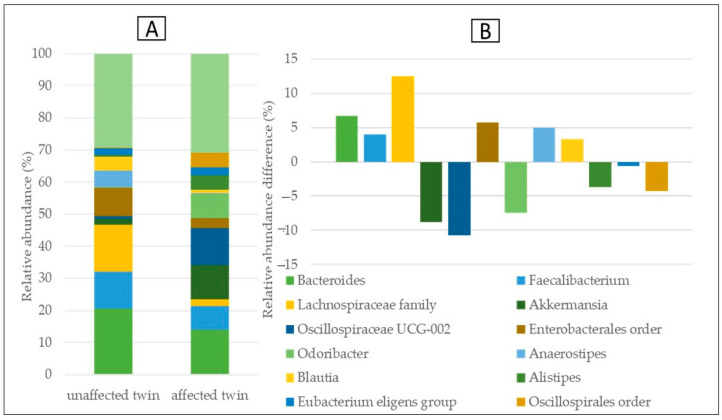
Relative abundances (**A**); the differences between abundances concerning the PSP discordant twin pair (**B**) at the genus level.

**Table 1 medicina-59-01696-t001:** Descriptive analysis and risk factors of the twin pair. Differences are shown with asterisk (*).

	Unaffected Twin	Affected Twin	Difference
Birth weight (g)	3500	3500	-
Sport activity throughout the life	Yes (Running)	No	*
Coffee consumption	Yes (1 cup/day)	No	*
Smoking	Never	Yes (21.5 packs a years)	*
Hypertension	Yes (40 years)	Yes (20 years)	
Cardiovascular disease	No	No	-
Diabetes	No	Yes (4 years)	*
Dyslipidemia	No	No	-
Cancer	No	Breast cancer	*
Year at menopause	55	60	*
Birth control pills	No	Yes (30 years)	*
COVID infection	No	Yes (In the 2nd wave in 2022)	*

**Table 2 medicina-59-01696-t002:** Clinical parameters and blood biomarkers of the twin pair.

	Unaffected Twin	Affected Twin
BMI (kg/m^2^)	29.4	21.7
Body fat (%)	38.4	30.5
Visceral fat scale (1–30)	12	7
Body muscle (%)	27.2	27.9
SBP (mmHg)	151	167
DBP (mmHg)	79	82
Total cholesterol (mmol/L)	5.1	4.9
LDL-C (mmol/L)	3.47	3.36
HDL-C (mmol/L)	1.55	1.30
Triglyceride (mmol/L)	1.1	1.2
ApoA1 (mg/dL)	1.79	1.5
ApoB (mg/dL)	1	0.98
Lp(a)	0.03	0.03
ApoE genotype	3/3	3/3
Apo-AII (mg/dL)	119.85	115.27
Apo-CII (mg/dL)	73.23	59.35
Apo-CIII (mg/dL)	17.50	15.72
Apo-D (mg/dL)	24.68	21.36
Apo-E (mg/dL)	71.26	61.65
Apo-H (mg/dL)	23.96	13.59
Apo-J (mg/dL)	45.27	42.14
Apo-M (mg/dL)	15.35	11.82
Aβ40 (pg/mL)	462.02	416.79
Aβ42 (pg/mL)	77.00	70.75
TAU (pg/mL)	48.79	36.83
NFL (pg/mL)	71.11	67.73
αSyn (pg/mL)	11.42	42.95
ICAM1 (ng/mL)	5.71	4.63
VCAM1 (ng/mL)	0.84	0.62
MMP2 (ng/mL)	76.08	36.59
MMP9 (ng/mL)	26.60	21.94
NGAL (ng/mL)	45.74	48.98
CysC (ng/mL)	162.33	130.66
MB (ng/mL)	42.09	10.48
MPO (ng/mL)	417.61	479.68
MRP8/14 (ng/mL)	2.62	2.51
OPN (ng/mL)	12.38	12.66
SAA (ng/mL)	45.75	46.74
MMP9/MMP2	0.3497	0.5997
Aβ42/Aβ40	0.1667	0.1697
TAU/Aβ42	0.6336	0.5205
ApoA1/SAA	0.5015	0.3150
ApoA1	22.00	14.00
ApoA1/SAA	4.8091	2.9950
ApoB/ApoA1	0.56	0.65
HDL/SAA	3.39	2.78
HDL/ApoD	6.28	6.09
HDL/ApoM	10.10	11.00
HDL/ApoJ	3.42	3.09

BMI: body mass index; SBP: systolic blood pressure; DBP: diastolic blood pressure; HDL-C: high-density lipoprotein cholesterol; LDL-C: low-density lipoprotein cholesterol; ApoA1: apolipoprotein AI, ApoB: apolipoprotein B; Lp(a): lipoprotein (a); ApoE: apolipoprotein E; Apo-AII: apolipoprotein AII; Apo-CII: apolipoprotein CII; Apo-CIII: apolipoprotein CIII; Apo-D: apolipoprotein D; Apo-H: apolipoprotein H; Apo-J: apolipoprotein J; Apo-M: apolipoprotein M; Aβ40: amyloid-40; Aβ42: β-amyloid-42; TAU: tau protein; NFL: neurofilament light chain; αSyn: α-synuclein; ICAM1: intercellular adhesion molecule 1; VCAM1: vascular cell adhesion molecule-1; MMP2: matrix metalloproteinase-2; MMP9: matrix metalloproteinase-9; NGAL: lipocalin A; CysC: cystatin C; MB: myoglobin; MPO: myeloperoxidase; MRP8/14: calprotectin; OPN: osteopontin; SAA: serum amyloid A.

**Table 3 medicina-59-01696-t003:** Cognitive examination and mental state tests. Remarkable differences are shown with asterisk (*).

		Unaffected Twin	Affected Twin
Handedness		Right	Right
Addenbrooke Cognitive Examination	Attention	18 points	13 points
	Memory	29 points	21 points
	Fluency	10 points	2 points
	Language	27 points	25 points
	Visuospatial skills	5 points	0 point
	Total points	89 points *	61 points *
Mini-Mental State Examination		30 points *	22 points *
Montreal Cognitive Assessment		28 points *	13 points *
Beck Depression Inventory		2 points	4 points
Geriatric Depression Scale		0 point	1 point
Zung Self-Rating Depression Scale		27 points	28 points

**Table 4 medicina-59-01696-t004:** White matter hyperintensity parameters of the twin pair.

Variable	Unaffected Twin	Affected Twin
Total lesions count	24	36
Total lesions volume (cm^3^)	3.9192	30.0650
Periventricular lesions count	9	8
Periventricular lesions volume (cm^3^)	3.3902	29.4052
Deep white lesions count	12	17
Deep white lesions volume (cm^3^)	0.3136	0.3823
Juxtacortical lesions count	2	8
Juxtacortical lesions volume (cm^3^)	0.0362	0.2238
Infratentorial cerebellar lesions count	1	2
Infratentorial cerebellar lesions volume (cm^3^)	0.1793	0.0403
Infratentorial medullary lesions count	0	1
Infratentorial medullary lesions volume (cm^3^)	0	0.0132

**Table 5 medicina-59-01696-t005:** Brain volumetric parameters of the twin pair.

Volumes	Unaffected Twin	Affected Twin
Total white matter volume (cm^3^)	403.59	387.12
Total gray matter volume (cm^3^)	664.13	623.45
Total brainstem volume (cm^3^)	20.34	16.14
Frontal lobe volume (cm^3^)	168.08	155.58
Temporal lobe volume (cm^3^)	103.83	103.51
Parietal lobe volume (cm^3^)	97.28	89.20
Occipital lobe volume (cm^3^)	80.16	76.79

**Table 6 medicina-59-01696-t006:** Cerebrospinal fluid (CSF) volume of the twin pairs.

CSF Volume	Unaffected Twin	Affected Twin
Total CSF volume (cm^3^)	220.89	272.12
Inferior lateral ventricle volume (cm^3^)	4	7.30
Lateral ventricle volume (cm^3^)	64.49	94.48
3rd ventricle volume (cm^3^)	2.93	3.85
4th ventricle volume (cm^3^)	3.55	5.27
External CSF volume (cm^3^)	145.92	161.23

## Data Availability

Data supporting the reported results can be obtained from the authors by request.

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
