# Peer review of "Analysis of Genetic and MRI Changes, Blood Markers, and Risk Factors in a Twin Pair Discordant of Progressive Supranuclear Palsy"

_medicina, 2023, doi:10.3390/medicina59101696_

Round 1

Reviewer 1 Report

In this paper, focus on progressive supranuclear palsy, the authors analyze the genetic and MRI changes, blood markers, and risk factors in a twin. The gut microbiome was also examined and some changes were observed. Overall, the research topic is interesting; the manuscript is well-designed and written. Minor revision is requested before acceptance.

1.      Please make it constant, Apo A1 or Apo AI.

2.      How many times did you repeat for the blood sample collection and sequencing?

3.      For a better reading, please clarify the meaning of the different points in Table 3. The higher the better?

4.      It’s a good idea to use a twin pair for the study, one is affected and the other one is normal, if there is a significant change in terms of PSP in gene level, that will directly point out the factors; however, the sample amount is too small, it is recommended to also compare the general PSP patients with normal people to see what’s the difference.

5.      The authors should make the conclusions more profoundly.

Author Response

Answer to the comments of Reviewer 1:

  1. Please make it constant, Apo A1 or Apo AI.

Response: Thank you for your comment. As requested, we rephrased the words to Apo A1.

  1. How many times did you repeat for the blood sample collection and sequencing?

Response: Thank you for this comment. There was no repetition, both the blood sampling and the genotyping were done once, as the PCR reaction took place under appropriate controls, so no repetition was necessary. We added the following sentence to the Methods: „Therefore, no repeats of genotyping were needed”.

  1. For a better reading, please clarify the meaning of the different points in Table 3. The higher the better?

Response: We apologize for the misunderstanding. We added the following interpretation to the Methods related to Table 3: “Considering ACE, MMSE, MoCA there are different cut-off scores in the literature for cognitive impairment, however, the higher the points, the better. If MMSE score is below 24 points, the result is frequently thought to be abnormal. For ACE 88 and 83 cut-off scores are recommended usually. If we find that the score of the MoCA test is below 25, a cognitive impairment might be present.  The cut-off score for BDI is 9, for GDS is 5 and for ZDS is 41.”

  1. It’s a good idea to use a twin pair for the study, one is affected and the other one is normal, if there is a significant change in terms of PSP in gene level, that will directly point out the factors; however, the sample amount is too small, it is recommended to also compare the general PSP patients with normal people to see what’s the difference.

Response: We agree with your comment. We added the following sentence to the Limitations: “Though we got similar data (especially observing the cognitive tests and MRI results) in comparison to the general population, healthy and affected as well, our sample amount was small, further investigations are important.”

  1. The authors should make the conclusions more profoundly.

Response: According tot he request, we rephrased the Conclusions.

Reviewer 2 Report

Dear authors, this is an interesting and original work about the PSP that is still a challenge in a movement  disorders diagnosis and therapy.

Despite in your paper you describe only one case of twin one of which affected by PSP, the extreme rarity  of the disease, justify it.

Probably with one case reported from you, draw conclusions is hasty, and some results are not very believeble.

However, I appreciated this work. The introduction and conclusion are easy to read and clear. The figure are well presented and the methods also.

The bibliography are updated and consistent with the paper text

Author Response

Answer to the comments of Reviewer 2:

  1. Dear authors, this is an interesting and original work about the PSP that is still a challenge in a movement disorders diagnosis and therapy. Despite in your paper you describe only one case of twin one of which affected by PSP, the extreme rarity of the disease, justify it. Probably with one case reported from you, draw conclusions is hasty, and some results are not very believeble. However, I appreciated this work. The introduction and conclusion are easy to read and clear. The figure are well presented and the methods also.

The bibliography are updated and consistent with the paper text

Response: Thank you for your positive feedback. As requested, to further clarify the manuscript, we rephrased the manuscript and emphasized the limitations in the Discussion.

We hope that the above answers and explanations sufficiently address the criticism raised during the review process and that our revised manuscript will be acceptable for publication in Medicina journal.

Kind regards